# Impact of the COVID-19 Pandemic on Influenza Vaccination Coverage of Healthcare Personnel in Alicante, Spain

**DOI:** 10.3390/vaccines12040370

**Published:** 2024-04-01

**Authors:** María Guerrero-Soler, Paula Gras-Valenti, Guillermo Platas-Abenza, José Sánchez-Payá, Ángela Sanjuan-Quiles, Pablo Chico-Sánchez

**Affiliations:** 1Epidemiology Unit, Preventive Medicine Service, Dr. Balmis General University Hospital, Alicante Institute for Health and Biomedical Research (ISABIAL), 03010 Alicante, Spain; guerrero_marsol@gva.es (M.G.-S.); platas_gui@gva.es (G.P.-A.); sanchez_jos@gva.es (J.S.-P.); chico_pab@gva.es (P.C.-S.); 2Department of Community Nursing, Preventive Medicine and Public Health and History of Science, University of Alicante, 03690 Alicante, Spain; 3Department of Nursing, University of Alicante, 03690 Alicante, Spain; angela.sanjuan@ua.es

**Keywords:** healthcare personnel, influenza vaccine, vaccine coverage, infection associated with health care

## Abstract

Influenza is a health problem and vaccination is the most effective measure to prevent it. The objective of this study was to evaluate the impact of the COVID-19 pandemic on vaccination coverage (VC) against influenza in healthcare workers (HCWs). A cross-sectional study was conducted at the Dr. Balmis University General Hospital in the province of Alicante (Spain), in which vaccination data were collected retrospectively. Adverse effects (AEs) were detected via telephone call between 15 and 30 days after vaccination. The existence of significant changes in VC between the different seasons studied was evaluated using Chi square with a statistical significance level of *p* < 0.05. A total of 8403 HCWs vaccinated throughout the different seasons were studied. The vaccination coverage of HCWs for influenza pre-COVID-19 pandemic (2019/20 season) was 51.9%; increased during the pandemic to 67.9% (2020/21 season) and 65.5% (2021/22 season); and, after the pandemic, it decreased to 42.7% (2022/23 season) (*p* < 0.05). The most frequent reason for vaccination during the periods evaluated was “self-protection”, followed by “protection of patients” and “protection of family members”. Of all HCWs evaluated, 26.6% (1460/5493) reported at least one AE. During the COVID-19 pandemic, HCWs’ influenza vaccination coverage fluctuated considerably. There has been an increase in VC during the most critical moments of the pandemic, both in the 2020/21 and 2021/22 seasons, which has, subsequently, decreased in the 2022/2023 season, to levels below pre-pandemic (2019/2020 season), which justifies implementing specific measures to recover VC in Spain.

## 1. Introduction

Influenza is a respiratory disease caused by the influenza virus. It is a significant global public health issue, affecting 5% to 15% of the population annually [1,2,3]. According to the World Health Organization (WHO), seasonal flu epidemics result in 3 to 5 million cases of severe illness and 290,000 to 650,000 deaths worldwide each year [3]. These numbers result in prolonged hospital stays and, consequently, increasing healthcare costs [4].

Vaccination is the most effective measure against influenza and its complications [5]. Vaccination programs have focused on seasonal coverage for individuals aged 65 and over or those who are immunocompromised [3]. Since 1980, the World Health Organization (WHO), the Centers for Disease Control and Prevention (CDC), and the European Union’s Committee on Safety have considered healthcare workers (HCWs) as a priority group for receiving the influenza vaccine [6,7,8]. The reasons for influenza vaccination among this group are self-protection; avoiding transmission to patients, colleagues, and the community; and simultaneously reducing absenteeism rates [7,8,9,10,11,12,13,14,15,16]. Additionally, HCWs, by getting vaccinated, set an example for the general population, thereby contributing to increasing vaccination coverage (VC) among patients.

Some influenza virus-induced pandemics have occurred, with the most recent following the emergence of human infection cases involving a new influenza virus (H1N1) in April 2009 [17,18]. This event marked a pivotal moment in the distribution of vaccination coverage among various healthcare professional categories [19]. Subsequently, in March 2020, health authorities declared the COVID-19 pandemic, caused by the SARS-CoV-2 virus, which leads to severe acute respiratory syndrome.

To further improve our understanding of HCWs’ vaccination rates, it is crucial to consider additional factors [20,21] such as the accessibility and convenience of vaccination services within healthcare facilities, educational efforts to correct safety misinformation and efficacy of vaccines, institutional policies regarding vaccination, personal beliefs, and attitudes toward vaccination, previous experiences with vaccines, and the impact of workload and job stress on vaccination decisions.

Historically, healthcare workers have been reticent to flu vaccination [6,7,8], with global vaccine coverage levels varying widely between 5% and 90% [7,20], and in Spain between 20% and 60% [19,21], but the COVID-19 pandemic has heightened awareness of its importance [22]. It is crucial to analyze the VC on healthcare workers (HCWs) and their evolution in the current context, particularly after the acute phase of COVID-19 [23], in order to understand the main factors driving HCWs towards vaccination. Furthermore, this analysis will enable us to assess the real impact of the COVID-19 pandemic on different seasons (during and after the pandemic) and to identify differences compared to previous periods. By doing so, we can establish a baseline for developing new strategies focused on improving flu vaccination campaigns in the long term, with the primary objective of enhancing their effectiveness.

On the other hand, examining one of the ongoing reasons for healthcare workers’ reluctance, such as potential adverse events (AEs), will furnish us with valuable insights to enact the necessary improvements.

In this context, the primary aim of this study was to outline the vaccination coverage (VC) trends among healthcare workers in a tertiary-level hospital from the 2019/2020 to the 2022/2023 seasons. Additionally, the secondary objective was to investigate potential variations in the factors influencing healthcare workers’ acceptance of influenza vaccination and the occurrence rates of adverse events (AEs) across these seasons.

## 2. Materials and Methods

### 2.1. Study Design, Participants, and Data Collection

A cross-sectional study was carried out at the Doctor Balmis University General Hospital in Alicante, Spain, which covered the influenza seasons from 2019/2020 to 2022/2023, whose data were obtained retrospectively.

Vaccination information was retrieved from the Nominal Vaccination Registry (NVR), a database that records all vaccination procedures in real-time, performed by health professionals as part of their daily routine in the Valencian Community. For this study, we specifically collected the vaccination records of healthcare staff contractually linked to the hospital during the relevant influenza seasons, who were eligible for vaccination.

The hospital’s influenza vaccination campaign usually starts in late October and ends in early February. Vaccinations and their corresponding registration in the NVR are carried out by the nursing staff of the Preventive Medicine Service, following the Influenza Vaccination Program protocol. Before each campaign begins, promotional efforts are made to increase vaccine acceptance among healthcare workers.

Furthermore, through the center’s established vaccination program, a survey is routinely conducted to understand the motivation of healthcare staff for getting vaccinated at the time of vaccination (added as Appendix A). This survey includes items such as gender, professional category, date of birth, and a list of reasons for accepting vaccination (self-protection, it is my duty, I believe it is advisable, it is free, I have been vaccinated before, to protect patients, to protect my family, history of having had flu, having a chronic illness, living with people over 65, recommended by a doctor). Healthcare professionals submit this survey at the time of vaccination.

As part of the same program, monitoring of adverse effects is carried out through a phone call between 15 and 30 days after vaccination. Through the telephone call, the health professional reports any possible adverse effects of the vaccine, by answering questions that are asked in a direct manner. Adverse effects were reported using clinical criteria already defined as the presence of any of the following symptoms within 2–3 days after vaccine administration: pain (discomfort at or around the injection site), malaise (feeling of physical discomfort), myalgia (generalized muscle discomfort), fever (temperature above 37.5 °C), and others (any other adverse reaction not among the previous ones).

Data for this study were retrospectively compiled from various databases used over the years by the center’s influenza vaccination program. The analysis of this data was authorized after obtaining approval from the ethics committee in 2023, when it was decided that all available information for analysis would be compiled.

### 2.2. Statistical Analyses

The 2019/2020 season was established as a pre-pandemic reference period preceding the COVID-19 pandemic, whereas the 2020/2021 and 2021/2022 seasons were designated as the pandemic period. Thereafter, the 2022/2023 season was identified as the post-pandemic period. A descriptive study of all the variables was carried out and differences in vaccination coverage (VC) were analyzed depending on the studied season, age, sex, and professional category. The VC for each season was calculated using the following formula: VC = number of HCWs vaccinated in each season/number of HCWs contractually linked to the center × 100 in each season. Afterwards, an analysis was conducted to detect differences between the reasons for vaccination acceptance and the occurrence of AEs (type and number of AE) depending on the study period. The chi-square test was used for the association study, considering *p* < 0.05 as statistically significant. The statistical analysis was performed using IBM^®^ SPSS^®^ Statistics v.25.0.

### 2.3. Ethics Statement

The present study was conducted in accordance with the principles embodied in the Declaration of Helsinki. This study was reviewed and approved by the Drug Research Ethics Committee at the Department of Health of the Hospital Dr. Balmis, in the city of Alicante (Ref.: 2023-115).

## 3. Results

Influenza vaccination coverage (VC) among HCWs at Alicante General University Hospital Dr. Balmis before the COVID-19 pandemic (2019/20 season) was 51.9% (1599/3079) and increased during the pandemic in the 2020/21 and 2021/22 seasons to 67.9% and 65.5% (2505/3827), respectively. However, VC saw a decline to 42.7% (1723/4036) in the post-acute phase of the pandemic (2022/23 season). These variations in VC across different time periods were statistically significant (*p* < 0.001) (Table 1).

Differences in the distribution of characteristics among vaccinated healthcare workers were observed across seasons (Table 2). In the 2019/20 season, 51.3% (820/1599) were aged 45 years or older, whereas this proportion increased to 60.8% (1048/1723) in the 2022/23 season. Regarding professional roles, in the pre-pandemic 2019/2020 season, 31.4% (502/1599) were physicians and 17.8% (284/1599) belonged to other categories. However, during the subsequent 2020/2021 season (during the pandemic), the proportion of physicians decreased to 20.5% (527/2576), while the “other” category comprised 27.7% (714/2576).

The primary reason for vaccination across all seasons was “self-protection” with frequencies ranging from 70.9% (1827/2576) in the 2020/21 season to 76.5% (1916/2505) in the 2021/22 season. Following this, “patient protection” peaked at 64.6% (1025/1599) in the 2019/20 season, while “protection of family members” reached 62.5% (1610/2576) during the same season (Table 2).

Among the total of 8403 healthcare workers (HCWs) vaccinated throughout the various seasons under study, 2505 were unable to undergo assessment for adverse effects during the 2021/22 season, since resources were focused on the COVID-19 pandemic. The assessment of adverse effects (AEs) was conducted in 93.1% (5493/5898) of cases, with 26.6% (1460/5493) experiencing at least one AE. The most frequently reported AE across all seasons was injection site pain, affecting 22.4% (310/1381), followed by discomfort at 12.3% (170/1381). The highest AE rate occurred during the 2022/23 season, with 32.2% (445/1381) of vaccinated healthcare workers (HCWs) experiencing AEs. Other AEs observed in HCWs ranged from 4.3% (110/2554) to 7.8% (121/1558), with no reports of allergic reactions or severe AEs (Table 2).

Finally, during the 2022/23 season, no significant association was observed between age (*p* = 0.073), gender (*p* = 0.070), professional category (*p* = 0.167), and previous vaccination (*p* = 0.761) with the occurrence of adverse effects (AEs) (Table 3).

It was observed that “self-protection” was one of the reasons associated with vaccination among healthcare workers (HCWs) aged 45 years or older (*p* < 0.001). Furthermore, “self-protection” was the most frequently reported motivation for vaccination across all categories of HCWs during the 2022/23 season, with frequencies of 76.1% (407/535) among physicians, 71.2% (327/459) among nursing staff, 75.9% (268/353) among auxiliary/technical staff, and 73.4% (276/376) among other HCWs (Table 3).

Conversely, having received prior vaccination was independently and significantly associated with all vaccination motives. Among the total vaccinated healthcare workers (HCWs), 85.0% (600/706) had a history of previous vaccination for “self-protection”; 59.9% (423/706) because they “consider it their duty”; 68.4% (483/706) because they believe “it is advisable”; 78.9% (557/706) to “protect the health of their family”; and 74.2% (524/706) to “protect the health of their patients” (Table 3).

## 4. Discussion

The data presented in this paper illustrate the positive impact of the acute phase of the COVID-19 pandemic on increasing influenza vaccination coverage (VC) among healthcare workers (HCWs) at Hospital General Universitario Dr. Balmis. This impact led to achieving a VC close to 70%, marking the historically highest recorded VC. Unfortunately, this increase was not sustained over time and, in the immediately following season (2022–2023 season), a significant decline was observed, falling below even the pre-pandemic levels.

During the pandemic, several studies [5,22,24,25,26,27,28,29] reported a similar increase in VC, with figures around 60–70%, which resemble the data obtained in our study during the 2020/21 and 2021/2022 seasons, at 67.9% and 65.5%, respectively. Nevertheless, the main finding of this study is the low VC for influenza observed in HCWs during the 2022/23 season (post-pandemic), at 42.7%, which is much lower than the pre-pandemic levels of 51.9% (2019/2020 season). This data aligns with an analysis conducted in intensive care units and nursing homes on HCWs’ VC [30], which also observed a marked decrease in VC. Some studies had already indicated a certain decline in VC during the 2021/2022 season [5,24,27,30,31,32,33,34,35], but the figures still remained similar to those obtained at the beginning of the pandemic, around 50–60%.

Both the observed data and the fact that, despite the increase in VC during the pandemic, it did not reach the 75% recommended by the World Health Organization [20], have led us to reflect on the reasons why healthcare workers (HCWs) choose to vaccinate themselves and find ways to improve these VC figures.

According to the results, we can observe how “self-protection” continues to be the main motivation for influenza vaccination among healthcare workers (HCWs) in all seasons studied, followed by “protecting the health of patients” and “protecting the health of family members”. Data published by other previous authors [22] and during the pandemic [2,34,36,37] also reflect the same motivations for HCWs’ vaccination.

In the literature, we can observe that some of the main reasons for rejecting vaccination are as follows: the lack of a sense of influenza risk and the belief that the flu is not a severe illness [2,27,34,38,39,40,41], concerns about side effects [27,32,34,35,36,37,40,41,42,43], distrust of vaccine safety [22,32,34,35,37,39], doubts about vaccine efficacy [27,32,34,35,37,39,40], considering it unnecessary [32], self-perception as a healthy population [22,27,36], and workload as a barrier to either accessing vaccination points or a lack of time [27,32,39,44].

Therefore, given that the main reason for vaccination is “self-protection,” there is a high percentage of individuals not only lacking a sense of risk, but who also perceive themselves as part of a healthy population and downplay the severity of the flu, which means that we are missing the opportunity to vaccinate many professionals. During the 2020/21 and 2021/22 seasons, the perception of risk was increased by the COVID-19 pandemic, with constant information about the importance of vaccination, the number of hospital admissions due to respiratory infections, and accessibility to rapid diagnostic tests, among other factors. These circumstances might have influenced the observed vaccine coverage during those seasons.

Furthermore, the fact that in the 2021/22 and 2022/23 seasons, the co-administration of the COVID-19 vaccine with the flu vaccine has been carried out in most centers may have increased distrust towards the safety of the vaccine and concern about its side effects. Consequently, this could have negatively influenced vaccination rates, especially during the 2022/23 season. Adding to this, rejection due to doubts about the vaccine’s efficacy presents a broad scope for improvement, where training could help reverse the vaccine-induced distrust.

Moreover, the co-administration of the COVID-19 vaccine with the flu vaccine in most centers during the 2021/22 and 2022/23 seasons may have heightened distrust regarding the vaccine’s safety and raised concerns about its potential side effects. As a result, this could have had a detrimental impact on vaccination rates, particularly during the 2022/23 season. Additionally, rejection stemming from doubts about the vaccine’s efficacy offers significant room for improvement, suggesting that training could play a crucial role in addressing vaccine-induced distrust.

Through the results, we can see that, of the total vaccinated, around 70–75% are women and between 25 and 30% are men. This observation is seen throughout all the seasons studied (Table 2), something that coincides with the fact that two-thirds of the professionals contractually linked to the hospital during the relevant influenza seasons were women. But, if we look at vaccination coverage, we can see that the figures between both sexes do not show great differences between the different seasons (Table 1).

In addition, it has been observed that medical and nursing staff normally have a higher vaccine coverage (VC) compared to other professional categories [5,27,28,35,36,37,42,43,45]. However, during the 2020/21 season, our study noted an increase in vaccine coverage among the “other” professional category. This coincided with the proliferation of information provided by the median and an increased perception of risk among the population. A similar situation was observed in a study conducted in 2012 [19], which evaluated the influenza vaccination in HCWs following the H1N1 flu pandemic. Additionally, it has also been perceived that, in both 2021 and 2022, other professional categories showed a greater intention to get vaccinated [2]. These data demonstrate the influence that information and training can have on shaping one’s perception of risk, which influences VC, especially when the primary motivation for vaccination is self-protection. There is a clear need to enhance training among healthcare workers (HCWs) and to implement vaccination campaigns with the continuous and active dissemination of information [45]. The implementation in healthcare student classrooms and the training provided to HCWs before beginning internships in healthcare centers represent a fundamental pillar. It will help raise awareness of the importance of vaccination and sensitize them to the role of healthcare workers as both recipients and transmitters of infections [4,44,46].

In the same vein, free vaccination has been associated with a higher degree of compliance with vaccination programs [42,44,47], thereby addressing inequalities in access. In places where influenza vaccination is not free, this could be a measure to consider. Conversely, incentive programs based on financial compensation to healthcare professionals for getting vaccinated have shown less success [48], even generating controversy in this regard [40].

Similarly, the mandatory nature of vaccination has led to differing opinions. While authors have observed high and sustained vaccination rates through mandatory vaccination policies [24,32,35], the demonstrated efficacy is limited and sparks an open debate [48]. This raises doubts about the ethics of demanding and enforcing such policies, adding to the discomfort caused by the lack of autonomy and loss of empowerment experienced by HCWs, significantly impacting their decision-making ability and morale [24,35,42,46,48]. Closely related, it has been shown that the perceived pressure for HCWs to get vaccinated turned out to be a significant barrier in many cases during the COVID-19 pandemic [49]. To address issues like this, some authors advocate for the use of “refusal statements”, in which healthcare professionals (HCWs) outline the reasons why they choose not to get vaccinated. This approach allows for the identification of the most significant barriers to vaccination success [47]. Other authors suggest the request for exemption from vaccination, followed by a critical evaluation of knowledge, attitudes, and practices regarding vaccination [31,50]. This allows for the implementation of much more focused information addressing the concerns of HCWs and a much more effective resolution aimed at promoting greater vaccine acceptance [31,50].

Another factor that influences HCWs when it comes to vaccination is the “fear of adverse effects (AEs)”. Among these, local pain stands out as the main AE, data that coincide with those reported in the previous literature [51,52]. We must take into account that co-administration with the COVID-19 vaccine increased AE incidence [51], from 15 to 20% in our study. This difference may be due to a difference in reactogenicity compared to separate vaccine administration. Nonetheless, pain remained the main AE found. It is crucial to consider this, as it may have impacted HCWs’ decision not to get vaccinated. More personalized vaccination strategies could help us reach all HCWs and may also be useful in reducing this fear of AEs [30]. Establishing personal coaching strategies aimed at promoting influenza vaccination among HCWs through media, building professional networks, and addressing HCWs’ doubts and concerns that may hinder vaccination [25] could be helpful.

The results obtained must be carefully analyzed, as we have seen they are influenced by multiple factors, and different characteristics are present depending on the healthcare policies of the country. The vaccination of HCWs can be influenced by various factors, including vaccine accessibility. Thus, reducing barriers to vaccination and implementing active influenza vaccination campaigns among HCWs undoubtedly contribute to increased vaccination rates [4,25,32,47]. Active vaccination strategies [32,35,37,42,43,44,45,51] that have proven to be effective include free vaccination, vaccine availability, active search of HCWs for on-site vaccination during working hours, resolution of doubts along with information dissemination, and flexibility of vaccination schedules at a fixed vaccination point. Nevertheless, despite implementing several strategies to improve vaccination rates post-pandemic, we observe that the results have not been satisfactory. Additional efforts will be needed to increase influenza vaccination coverage among HCWs, considering the influence of information sources, particularly social media [34,37,43,44]. While these platforms can be of great assistance, they can also have a negative impact on vaccination campaigns by disseminating misinformation or inaccurate information.

It is essential that political bodies and public health organizations present specific initiatives to increase confidence in vaccines and launch public vaccination campaigns raising awareness about the importance of vaccination and their safety, taking into account sociocultural factors. Otherwise, they could have a negative influence on vaccination coverage [38,42,43].

### Limitations

Among the limitations of these findings, it is possible that some vaccines are not registered in the Vaccination Registry of the Valencian Community, which could lead to an underestimation of vaccine coverage (CV). This can occur in those cases of health professionals who have not been vaccinated by the previously trained nursing staff of the Preventive Medicine service. Some health professionals do not have training when registering vaccines in the NRV and this may lead to an underestimation of vaccine coverage in some cases. To mitigate this, vaccines for health professionals at our hospital have been administered and registered in the NRV by nursing staff from the center’s Preventive Medicine Service with prior training in this regard.

However, it is important to note that not all adverse event (AE) data could be obtained from vaccinated healthcare professionals, due to the interruption of telephone follow-up for two seasons, as human resources were diverted to tasks related to the COVID-19 pandemic. Furthermore, because AEs were collected retrospectively between 15 and 30 days after vaccination, it was not feasible to control for bias arising from each professional’s memory. Nevertheless, efforts were made to minimize this bias by specifically inquiring about common symptoms, as well as less common ones.

This study may be inherently limited by health workers’ self-reported vaccination motivations, as it may be influenced by their inclination to provide socially desirable responses. This can happen, especially in a hospital setting, where vaccination may be seen as a professional responsibility. Moreover, the preceding vaccination promotion campaign may have influenced the influenza vaccination results. However, these effects remained constant across all observations in the study.

Furthermore, the study spans several influenza seasons, which introduces variables beyond our control, such as changes in vaccine composition and the prevalence of viral strains. Additionally, factors directly related to our center, like yearly variations in vaccination practices, could also influence the results. These temporal changes may complicate the interpretation and applicability of our findings over time. To minimize their impact, we have consistently followed the vaccination protocols established by our center’s Preventive Medicine Service with minimal changes over the years.

## 5. Conclusions

During the COVID-19 pandemic, influenza vaccination coverage (VC) among healthcare workers (HCWs) has varied considerably. There was an increase in VC during the most critical moments of the pandemic, in the 2020/21 and 2021/22 seasons, followed by a decrease in VC in the 2022/2023 season to levels lower than those in the seasons before the pandemic (2019/2020 season).

It is crucial to monitor this trend in influenza VC among HCWs and conduct continuous monitoring, as it is a fundamental pillar to prevent nosocomial transmission of the influenza virus. Evaluating and analyzing the factors influencing HCWs’ willingness to receive the vaccine will enable us to implement improvement strategies focused on increasing VC. The results obtained indicate that there is now more room for improvement than ever before. Efforts should be focused on designing vaccination campaigns that take into account HCWs’ motivations for vaccination. Additionally, it is imperative to delve into the reasons behind HCWs’ decisions not to get vaccinated, with the primary objective of increasing VC in this group, whose role is paramount in preventing disease transmission and setting an example for the population to follow.

## Figures and Tables

**Table 1 vaccines-12-00370-t001:** Flu vaccination coverage among healthcare personnel classified by age, gender, and professional category according to the time period.

	COVID-19 Pre-Pandemic	During the Pandemic		COVID-19 Post-Pandemic	p1*	p2*	p3*	p4*	p5*	p6*
	2019–2020	2020–2021	2021–2022	2022–2023						
**Total**	51.9% (1599/3079)	67.9% (2576/3796)	65.5% (2505/3827)	42.7% (1723/4036)	<0.001	<0.001	0.028	<0.001	<0.001	<0.001
**Age**										
<45	60.7% (779/1283)	87.4% (1285/1470)	68.2% (1146/1681)	29.9% (675/2256)	<0.001	<0.001	<0.001	<0.001	<0.001	<0.001
≥45	45.7% (820/1796)	55.5% (1291/2326)	63.3% (1359/2146)	58.9% (1048/1780)	<0.001	<0.001	<0.001	<0.001	0.036	0.004
**Gender**										
Male	52.0% (426/819)	71.7% (681/950)	69.2% (695/1004)	48.8% (521/1068)	<0.001	<0.001	0.254	0.179	<0.001	<0.001
Female	51.9% (1173/2260)	66,6% (1895/2846)	64.1% (1810/2823)	40.5% (1202/2968)	<0.001	<0.001	0.054	<0.001	<0.001	<0.001
**Professional category**										
Physicians	67.2% (502/747)	66.5% (527/793)	74.4% (625/840)	56.7% (535/943)	0.798	0.002	<0.001	<0.001	<0.001	<0.001
Nurses	55.6% (533/958)	69.8% (854/1224)	60.1% (716/1192)	35.8% (459/1283)	<0.001	0.043	<0.001	<0.001	<0.001	<0.001
Nurse’s aide/ technician	34.8% (280/805)	45.5% (481/1058)	55.5% (571/1029)	33.6% (353/1052)	<0.001	<0.001	<0.001	0.615	<0.001	<0.001
Other ^1^	49.9% (284/569)	99.0% (714/721)	77.4% (593/766)	49.6% (376/758)	<0.001	<0.001	<0.001	0.956	<0.001	<0.001

^1^ Other: hospital porter, maintenance staff, administrative staff, cleaning, etc. p1*: comparison 2020–2021/2019–2020. p2*: comparison 2021–2022/2019–2020. p3*: comparison 2021–2022/2020–2021. p4*: comparison 2022–2023/2019–2020. p5*: comparison 2022–2023/2020–2021. p6*: comparison 2022–2023/2021–2022.

**Table 2 vaccines-12-00370-t002:** Characteristics of healthcare personnel, reasons for being vaccinated, and frequency of adverse events (AEs) according to the time period.

	Pre-Pandemic	During the Pandemic		Post-Pandemic	p1*	p2*	p3*	p4*	p5*	p6*
	2019–2020	2020–2021	2021–2022	2022–2023						
**Number vaccinated**	1599	2576	2505	1723						
**Age**										
<45	48.7% (779)	49.9% (1285)	45.7% (1146)	39.2% (675)	0.464	0.063	0.003	<0.001	<0.001	<0.001
≥45	51.3% (820)	50.1% (1291)	54.3% (1359)	60.8% (1048)						
**Gender**										
Male	26.6% (426)	26.4% (681)	27.7% (695)	30.2% (521)	0.884	0.439	0.294	0.022	0.006	0.078
Female	73.4% (1173)	73.6% (1895)	72.3% (1810)	69.8% (1202)						
**Professional category**										
Physicians	31.4% (502)	20.5% (527)	25.0% (625)	31.1% (535)	<0.001	<0.001	<0.001	<0.001	<0.001	<0.001
Nurses	33.3% (533)	33.2% (854)	28.6% (716)	26.6% (459)						
Nurse’s aide/technician	17.5% (280)	18.7% (481)	22.8% (571)	20.5% (353)						
Other ^1^	17.8% (284)	27.7% (714)	23.7% (593)	21.8% (376)						
**Reasons for being vaccinated**										
Protect my health	71.8% (1138)	70.9% (1827)	76.5% (1916)	74.2% (1278)	0.566	0.001	<0.001	0.117	0.020	0.085
It is my obligation	50.3% (804)	50.9% (1312)	51.7% (1296)	49.7% (857)	0.881	0.516	0.566	0.583	0.443	0.202
I think it is advisable	45.9% (728)	47.7% (1229)	46.4% (1163)	47.7% (822)	0.256	0.743	0.360	0.298	0.999	0.412
It is free	11.0% (175)	9.8% (253)	11.9% (298)	9.1% (156)	0.211	0.401	0.017	0.058	0.401	0.003
I was vaccinated previously	39.5% (626)	34.6% (892)	40.2% (1006)	41.0% (706)	0.002	0.672	<0.001	0.386	<0.001	0.596
I suffered flu in other years	10.8% (171)	8.2% (212)	8.0% (200)	11.5% (198)	0.006	0.002	0.748	0.517	<0.001	<0.001
Protect the health of my family	57.4% (910)	62.5% (1610)	57.5% (1440)	55.8% (962)	0.001	0.946	<0.001	0.371	<0.001	0.287
Protect the health of patients	64.6% (1025)	56.4% (1454)	55.1% (1381)	53.7% (925)	<0.001	<0.001	0.346	<0.001	0.075	0.354
Doctor’s recommendation	10.2% (162)	8.7% (225)	6.8% (170)	8.9% (154)	0.110	<0.001	0.010	0.212	0.818	0.010
I suffer a chronic disease	11.5% (183)	8.6% (222)	8.7% (217)	7.2% (124)	0.002	0.003	0.955	<0.001	0.093	0.085
I live with people >65	11.1% (176)	13.1% (337)	10.9% (274)	11.9% (205)	0.058	0.880	0.019	0.467	0.252	0.333
**Healthcare personnel evaluated**	97.4% (1558)	99.1% (2554)	NC ^2^	80.2% (1381)	<0.001	NC ^2^	NC ^2^	<0.001	<0.001	NC ^2^
**Healthcare personnel with AEs**	23.2% (361)	25.6% (654)	NC ^2^	32.2% (445)	0.079	NC ^2^	NC ^2^	<0.001	<0.001	NC ^2^
**Types of AEs ^3^**										
Pain	15.3% (238)	20.9% (533)	NC ^2^	22.4% (310)	<0.001	NC ^2^	NC ^2^	<0.001	0.249	NC ^2^
Discomfort	4.6% (72)	3.4% (88)	NC ^2^	12.3% (170)	0.059	NC ^2^	NC ^2^	<0.001	<0.001	NC ^2^
Myalgia	1.0% (15)	1.2% (30)	NC ^2^	6.5% (90)	0.526	NC ^2^	NC ^2^	<0.001	<0.001	NC ^2^
Fever	1.5% (24)	0.6% (16)	NC ^2^	4.6% (63)	0.004	NC ^2^	NC ^2^	<0.001	<0.001	NC ^2^
Other AEs ^4^	7.8% (121)	4.3% (110)	NC ^2^	7.2% (100)	<0.001	NC ^2^	NC ^2^	0.590	<0.001	NC ^2^
**Number of AEs ^3^**			NC ^2^							
One	75.3% (272)	83.3% (545)	NC ^2^	58.9% (262)	0.002	NC ^2^	NC ^2^	<0.001	<0.001	NC^2^
Two	20.2% (73)	15.0% (98)	NC ^2^	26.1% (116)						
Three or more	4.4% (16)	1.7% (11)	NC ^2^	15.1% (67)						

^1^ Other: hospital porter, maintenance staff, administrative staff, cleaning, etc. NC ^2^: Not calculated. ^3^ AE: Adverse events. ^4^ Other EA: Local redness, chills, local induration, sweating, etc. p1*: comparison 2020–2021/2019–2020. p2*: comparison 2021–2022/2019–2020. p3*: comparison 2021–2022/2020–2021. p4*: comparison 2022–2023/2019–2020. p5*: comparison 2022–2023/2020–2021. p6*: comparison 2022–2023/2021–2022.

**Table 3 vaccines-12-00370-t003:** Factors associated with the adverse events and the reasons for being vaccinated in the 2022–2023 season.

	Adverse Events	Protect My Health	It Is My Obligation	I Think It Is Advisable	Vaccinated Previously	Protect the Health of My Family	Protect the Health of Patients
	% (*n*) *p*	% (*n*) *p*	% (*n*) *p*	% (*n*) *p*	% (*n*) *p*	% (*n*) *p*	% (*n*) *p*
**Age**	0.073	<0.001	0.289	0.138	0.049	0.139	0.128
<45	35.1 (181/515)	69.5 (469/675)	48.1 (325/675)	45.5 (307/675)	38.1 (257/675)	53.6 (362/675)	51.4 (347/675)
≥45	30.5 (264/866)	77.2 (809/1048)	50.8 (532/1048)	49.1 (515/1048)	42.8 (449/1048)	57.3 (600/1048)	55.2 (578/1048)
**Gender**	0.070	0.166	0.742	0.491	0.871	0.842	0.975
Male	28.7 (119/414)	31.5 (398/521)	49.1 (256/521)	46.4 (242/521)	41.3 (215/521)	55.5 (289/521)	53.7 (280/521)
Female	33.7 (326/967)	31.5 (880/1202)	50.0 (601/1202)	48.3 (580/1202)	40.8 (491/1202)	56.0 (673/1202)	53.7 (645/1202)
**Professional category**	0.167	0.290	0.720	0.180	0.413	0.235	0.151
Physicians	31.3 (132/422)	76.1 (407/535)	50.7 (271/535)	48.0 (257/535)	39.8 (213/535)	55.1 (295/535)	53.1 (284/535)
Nurses	34.3 (123/359)	71.2 (327/459)	51.0 (234/459)	44.7 (205/459)	38.8 (178/459)	54.0 (248/459)	50.3 (231/459)
Nurse’s aide/technician	35.6 (106/298)	75.9 (268/353)	47.3 (167/353)	46.5 (164/353)	42.5 (150/353)	60.6 (214/353)	58.4 (206/353)
Other ^1^	27.8 (84/302)	73.4 (276/376)	49.2 (185/376)	52.1 (196/376)	43.9 (165/376)	54.5 (205/376)	54.3 (204/376)
**Vaccinated previously**	0.761	<0.001	<0.001	<0.001	NC ^2^	<0.001	<0.001
Yes	31.8 (183/576)	85.0 (600/706)	59.9 (423/706)	68.4 (483/706)	NC ^2^	78.9 (557/706)	74.2 (524/706)
No	32.5 (262/805)	66.7 (678/1017)	42.7 (434/1017)	33.3 (339/1017)	NC ^2^	39.8 (405/1017)	39.4 (401/1017)

^1^ Other: hospital porter, maintenance staff, administrative staff, cleaning, etc. NC ^2^: Not calculated.

## Data Availability

Data that are not presented in the article are available upon reasonable request from the corresponding author.

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
