# Peer review of "Impact of the COVID-19 Pandemic on Influenza Vaccination Coverage of Healthcare Personnel in Alicante, Spain"

_vaccines, 2024, doi:10.3390/vaccines12040370_

Round 1

Reviewer 1 Report

Comments and Suggestions for Authors

The manuscript (ID: vaccines-2921836) aimed to evaluate the impact of the COVID-19 pandemic on vaccination-coverage against influenza in healthcare workers in Spain. 

Comments:    

Work as a whole:

  • References should be cited in the correct order in the text. Check whether the order of references in the text corresponds to the references in the References list. To correct.   

Introduction:

  • Line 65: Add a new paragraph in which to present in detail the coverage of influenza vaccination among healthcare workers during a long period before the pandemic. Present this data for Spain and other countries. 

Methods:

  • The section Methods must be substantially supplemented with data about `Study population`, `Study sample`, `Sample size calculation`, `Response rate`, eligibility criteria.  
  • Define the time when this study was conducted.  
  • Lines 73-75: If the statement in this sentence is correct, I quote `A cross-sectional study was conducted at the Hospital General Universitario Doctor Balmis in the province of Alicante during the influenza seasons from 2019/2020 to 2022/2023.`, explain how the approval of the Ethics Committee for this study was obtained only in 2023 (listed on Line 109 and Line 348).  

Results:

  • Results are clearly presented and described.   

Discussion:

  • In general, in the Discussion section, the results of this study are comprehensively discussed. The comparison of the results of this study with the results of similar studies was carried out in a qualitative manner.
  • Just one remark: It is necessary to add a consideration of the differences in the results by gender that are presented in this paper to the Discussion section. Give a possible explanation for the mentioned gender differences.        

Conclusions:

  • The conclusions are supported by the results.    

Reviewer 2 Report

Comments and Suggestions for Authors

 - Line 95. Please check the typo “2022/2021 season” (2022/2023?).

 - The authors should include more information about the questionnaire and how participants filled it in (time, where they filled out the questionnaire, how researchers collected it). Moreover, it would be useful for readers to include a copy (as supplementary files) of the questionnaire in the original language and in English.

- More information on the methodology used to detect adverse effects after administration of vaccination should also be included in the materials and methods. How did the authors verify that these were adverse events due to the administration of the vaccines? How were they defined? Were clinical criteria used? If so, the authors should better clarify these aspects.

- The authors should address more thoroughly in the Discussion section the limits regarding the study design.

Comments on the Quality of English Language

.

Reviewer 3 Report

Comments and Suggestions for Authors

•            MAJOR ISSUES

•            Institutional Review Board Statement: The study was conducted in accordance with the Declara- 346 tion of Helsinki and approved by the Drug Research Ethics Committee at the Department of Health 347 (PI2023/0328)

•            Please indicate the complete name of the Comitte  eg by the Drug Research Ethics Committee at the Department of Health  of the Hospital X in  the city, of  Alicante, or the University of Alicante, also include the date of approval (PI2023/0328)

•            Please after the name of the ethical comitte in English, write between  brackets and in itallics ithe origina  name of the the  committee in Spanish

•            In the Material and methdos  the authors said “vaccination status was retrospectively obtained 83 from the Nominal Vaccine Registry (RNV) of the Conselleria de Sanitat de la Generalitat 84 Valenciana at the end of each season.” , but later in the discussion  “Among the limitations of these findings, it is possible that some vaccinations are not 310 recorded in the Vaccination Registry of the Valencian Community, potentially resulting 311 in an underestimation of vaccination coverage (VC).”

•            The authors should describe the RNV, explain its coverage, included population, how data are collected etc.

•            Also in the discussion the authors said that even if there is subregistry , it is no problem because the vaccine were administered by the hospital nurses.  “Among the limitations of these findings, it is possible that some vaccinations are not 310 recorded in the Vaccination Registry of the Valencian Community, potentially resulting 311 in an underestimation of vaccination coverage (VC). To mitigate this, vaccinations have 312 been administered by the nursing staff of the Preventive Medicine Service at the centre (..)”   I don’t understand this, because if the source of the vaccination status is the RNV,  doesn`t  matter  who delivered the vaccine.

•            When providing proportion authors should include 95% confidence intervals.

•            In tables 1 and 2, the authors do multiple comparison of  pairs or dates  (p1,p2,p3,p4,p5,p6)  authors should do a correction for multiple comparisons such as Bonferroni.

Minor

•            In the introduction the authors deals with influenza , influenza vaccination, and the epidemic of covid 19.

•            There are other aspect that should be considered  What impact had

•            In the title, write the city and the country where the study was performed.

•            In the abstract include the institution, as well as the city and the country where the study was performed. Include the number of participant in the study.

•            In the last line of the abstract  change “ (…) VCs in our country” with “(…)VCs in  Spain”

•            Some times within the text, the authors included the name of the authors “While 258 authors such as Lim L et al, Shia X et al, and Kang M et al (…)To address issues like this, 266 authors such as Ajenjo et al (…), while other the references are included with a number. The system for the reference should be homogenous in the paper

Round 2

Reviewer 1 Report

Comments and Suggestions for Authors

Thank you for the opportunity to re-review the manuscript ID: vaccines-2921836. The authors responded to my comments. Thanks to the authors.

One note for the authors: the response in the `Cover Letter` states `(Added as supplementary material)`, but that note is missing from the text of the revised version of this paper. Need to check.   

If it is a Survey form, add it as a Supplementary file.   

Reviewer 3 Report

Comments and Suggestions for Authors

Authors have made most of my comments except about Bonferroni, but their explanation was satisfactory.